# Respiratory Syncytial Virus: New Challenges for Molecular Epidemiology Surveillance and Vaccination Strategy in Patients with ILI/SARI

**DOI:** 10.3390/vaccines9111334

**Published:** 2021-11-16

**Authors:** Fabio Tramuto, Carmelo Massimo Maida, Daniela Di Naro, Giulia Randazzo, Francesco Vitale, Vincenzo Restivo, Claudio Costantino, Emanuele Amodio, Alessandra Casuccio, Giorgio Graziano, Palmira Immordino, Walter Mazzucco

**Affiliations:** 1Department of Health Promotion Sciences Maternal and Infant Care, Internal Medicine and Medical Specialties “G. D’Alessandro”–Hygiene Section, University of Palermo, 90127 Palermo, Italy; carmelo.maida@unipa.it (C.M.M.); francesco.vitale@unipa.it (F.V.); vincenzo.restivo@unipa.it (V.R.); claudio.costantino01@unipa.it (C.C.); emanuele.amodio@unipa.it (E.A.); alessandra.casuccio@unipa.it (A.C.); palmira.immordino@unipa.it (P.I.); walter.mazzucco@unipa.it (W.M.); 2Regional Reference Laboratory for Molecular Surveillance of Influenza, Clinical Epidemiology Unit, University Hospital “Paolo Giaccone”, 90127 Palermo, Italy; dinaro.d@libero.it (D.D.N.); giulia.randazzo1986@gmail.com (G.R.); giorgio.graziano@policlinico.pa.it (G.G.)

**Keywords:** respiratory syncytial virus, molecular surveillance, risk factors, vaccination, community, hospitalization

## Abstract

Several respiratory pathogens are responsible for influenza-like illness (ILI) and severe respiratory infections (SARI), among which human respiratory syncytial virus (hRSV) represents one of the most common aetiologies. We analysed the hRSV prevalence among subjects with ILI or SARI during the five influenza seasons before the emergence of SARS-CoV-2 epidemic in Sicily (Italy). Respiratory specimens from ILI outpatients and SARI inpatients were collected in the framework of the Italian Network for the Influenza Surveillance and molecularly tested for hRSV-A and hRSV-B. Overall, 8.1% of patients resulted positive for hRSV. Prevalence peaked in the age-groups <5 years old (range: 17.6–19.1%) and ≥50 years old (range: 4.8–5.1%). While the two subgroups co-circulated throughout the study period, hRSV-B was slightly predominant over hRSV-A, except for the season 2019–2020 when hRSV-A strongly prevailed (82.9%). In the community setting, the distribution of hRSV subgroups was balanced (47.8% vs. 49.7% for hRSV-A and hRSV-B, respectively), while most infections identified in the hospital setting were caused by hRSV-B (69.5%); also, this latter one was more represented among hRSV cases with underlying diseases, as well as among those who developed a respiratory complication. The molecular surveillance of hRSV infections may provide a valuable insight into the epidemiological features of ILI/SARI. Our findings add new evidence to the existing knowledge on viral aetiology of ILI and SARI in support of public health strategies and may help to define high-risk categories that could benefit from currently available and future vaccines.

## 1. Introduction

Acute Respiratory Infections (ARIs) pose a relevant threat for public health because of their high morbidity and mortality, with upper and lower respiratory tract infections being primary causes of disease worldwide [1,2]. Contributing to more than 650,000 deaths in children younger than 5 years in 2016, they have a significant impact even among older adults [3,4].

ARIs were found to be associated with several different pathogenic bacterial and viral agents, among which are influenza virus and human respiratory syncytial virus (hRSV), although their aetiology often remains of unknown origin [5].

Almost every child by the age of 2 years gets infected with hRSV infection, usually with mild symptoms such as influenza-like illness (ILI). However, hRSV may also cause severe acute respiratory infection (SARI) that may lead to the involvement of the lower respiratory tract causing bronchiolitis, pneumonia, and tracheobronchitis [6,7].

SARI represents one of the prominent causes of hospitalization and death, especially among high-risk groups such as very young children, pregnant women, the elderly and individuals with underlying medical conditions [7,8].

In temperate northern hemisphere influenza and hRSV, through their types, subtypes, and subgroups, seasonal peaks during winter are shown, which generally overlap [9]. Even though viral co-circulation is a common phenomenon, each season can be characterized by different dynamics in molecular epidemiology, with the predominance of one virus type or subtype over another [10].

Of interest, it was found that the overall detection rates of the abovementioned viruses may vary disproportionately, according to the age of patients and healthcare setting [1]. In this regard, recent studies have recognized the importance of hRSV infections in adults, particularly individuals who are immunocompromised or have underlying lung or cardiovascular disease, as a cause of severe respiratory disease with complications, leading to prolonged hospital stay and high mortality rates [11].

Although the epidemiology of hRSV in high-income countries has been relatively well characterized, global surveillance programs have not been structured so far [12]. From the perspective to introduce new effective vaccines [13,14,15] and to address public health policies [16], it is of paramount importance to improve the knowledge on the molecular epidemiology of hRSV [17].

In Italy, an influenza surveillance system covering the whole country (InfluNet) has been operating over the last two decades [18]. Nevertheless, hRSV surveillance has not yet been fully implemented and the knowledge about spread and impact of this pathogen is mainly limited to data from individual research studies, primarily focused on children with influenza-like illness [19,20,21] or other specific settings [22].

To fill the knowledge gap, we conducted a retrospective cross-sectional study, within the activities of the Sicilian Regional Reference Laboratory (RRL), with the aim to investigate the molecular epidemiology and prevalence of hRSV among ILI outpatients and SARI inpatients of any age, identified by the InfluNet surveillance network in Sicily, the fifth most populated region of Italy, during the five influenza seasons before the emergence of SARS-CoV-2 epidemic.

## 2. Materials and Methods

### 2.1. Study Population and Inclusion Criteria

Patients meeting ILI or SARI case definitions [23,24] and residing in Sicily—the southernmost region of Italy accounting for nearly 5 million inhabitants out of 60 million at national level [25]—were included in the study.

A time-period encompassing five consecutive winter seasons (October to April), between 2015–2016 and 2019–2020, until the beginning of the COVID-19 pandemic, was considered.

General practitioners (GPs) and family paediatricians (FPs), contributing as sentinel physicians to the InfluNet network [26], consecutively collected oropharyngeal swabs from patients presenting ILI, while (oropharyngeal swabs or bronchoalveolar lavage-BAL, as appropriate) were sampled from patients admitted to hospital with SARI with the aim of evaluating the spread of hRSV in complicated cases.

All respiratory samples were conferred to the InfluNet Sicilian RRL, at the University Hospital “AOUP–P. Giaccone” of Palermo (Italy).

### 2.2. Laboratory Methods

Viral RNA was extracted using QIAamp Viral RNA extraction kit (QIAGEN, Hilden, Germany), according to the manufacturer’s suggested protocol, and RNA was then eluted from the spin column in 60 μL of elution buffer. Eluted RNA was divided into aliquots and stored immediately at −80 °C until further use.

Extraction performance was verified by means of a one-step real-time (rt)-PCR assay targeting the human ribonuclease P gene (RNase P) [27]. Samples showing a rt-PCR cycle threshold (Ct) value ≤35 were considered suitable to be tested for virus detection.

Singleplex one-step real-time retro-transcription (RT)-PCR assays were used to detect the presence of hRSV-A and/or hRSV-B [28,29]. A test was considered positive when its Ct value was <40.

All rt-PCR assays were performed with a QuantStudio™ 7 Flex Real-Time PCR System (Applied Biosystems, Carlsbad, CA, USA).

### 2.3. Data Collection and Analysis

At the moment of sample collection, physicians were asked to complete a structured form including the following information: date of sampling, patient’s initials, date of birth, sex, date of onset of symptoms, underlying medical conditions (such as cardiovascular disease, lung disease, diabetes, obesity, metabolic disorders, cancer, immunological disorders, genetic disorders, renal disease, neurological disease, liver disease, pancreatic disease), respiratory complications (pneumonia, respiratory failure, ARDS), and hospitalization (if required).

The study population was arbitrarily subdivided into nine different age-groups, five for children/teenagers (≤11 months, 12–23 months, 2–4, 5–10, and 11–18 years old) and four for adults/elderly (19–34, 35–49, 50–64, and ≥65 years old).

Descriptive statistics were used to summarize socio-demographic and clinical data of the recruited patients, as well as for viral characteristics. Median values and interquartile ranges (IQRs) were used to describe continuous variables, whereas frequency analyses for categorical variables were described with percentages. Comparisons of continuous variables were conducted using the Wilcoxon rank-sum test, according to data distribution, while the Pearson’s chi-square test was used for categorical variables. All the analyses with *p*-values of 0.05 or less were considered statistically significant (two tailed).

Time series data were taken into account in order to evaluate seasonal variations and periodic changes in prevalence of hRSV subgroups.

Moreover, the univariate logistic regression analysis was performed to identify the possible relationships between relevant clinical data (comorbidities and respiratory complications) and hRSV detection. Estimates were expressed as odds ratios (ORs) and/or adjusted odds ratios (adj-ORs), with their 95% confidence intervals (CIs).

Data were processed with the STATA MP statistical software package v16.1 for Apple™ (StataCorp LLC, College Station, TX, USA).

### 2.4. Ethical Review

This study was carried out in full compliance with the rules concerning the protection of personal data adopted in Italy and accomplished the Helsinki Declaration. All patients or their parents/legal guardians gave their consent for sampling and data collection, which were analysed anonymously.

The study was approved by the institutional ethics committee of the University Hospital “AOUP–P. Giaccone” of Palermo (Italy), approval number 09/2019, and was partially funded by Merck Sharp & Dohme Corp. The sponsor of the study had no role in study design, data collection, data analysis, data interpretation, or writing of the report. All authors had full access to all the data in the study and accepted responsibility to submit for publication.

## 3. Results

Table 1 shows the demographic and clinical characteristics of the 9584 subjects included in the study according to the surveillance in place. The ratio between males and females (M:F) was 1.05, and this data was substantially uniform across all surveillance seasons (range = 1.00–1.11, data not shown).

The median age was equal to 10 years old (IQR = 41) with children/adolescents (≤18 years old) accounting for 60.8% (*n* = 5831/9584) of the entire study sample.

Most of the participating subjects (84.8%; *n* = 8123/9584) were outpatients accessing community-based services and managed for ILI symptoms by “sentinel physicians”, whereas 15.2% (*n* = 1461/9584) were SARI patients admitted to hospitals. Of these, 29.8% (*n* = 435/1461) were patients admitted to ICUs.

The overall prevalence of hRSV was 8.1% (*n* = 772/9584), more than half of infections were caused by hRSV-B, while a small proportion of subjects (2.3%; *n* = 18/772) were co-infected with both subgroups of respiratory syncytial viruses. No difference was found by sex, neither in total nor by subgroups, whereas differences were found according to age. hRSV more frequently spread among children <5 years of age, ranging from 17.6% to 19.1%, the prevalence then rapidly decreased among adolescents and young adults, while the rate increased again up to 5.1% from the age of 50 years.

Of note, the median age of hRSV-A and hRSV-B cases significantly differed (*p* < 0.001). More in depth, children <5 years old and young adults (19–34 years old) generally were more infected by hRSV-A, while at least two-thirds of hRSV infections in individuals ≥35 years old were sustained by hRSV-B. All hRSV-A and B co-infected cases were children aged in the range between 12 months and 10 years old.

Overall, hRSV infections were 2-fold more frequent among outpatients when compared with subjects admitted to hospital (8.8% vs. 4.0%; *p*-value *<* 0.001) and, notably, the prevalence of infection differed appreciably in relation to hRSV subgroups.

Among community-based hRSV cases, usually presenting mild symptoms (ILI), no significant difference was found according to subgroups (47.8% and 49.7%, for hRSV-A and hRSV-B respectively), whereas among hospitalized patients, hRSV-B was clearly predominant over the other (69.5% vs. 30.5%), and this trend was even more appreciable among patients with severe clinical outcomes, which required an admission to the ICU (78.6%). Co-infections with both hRSV subgroups were exclusively documented in cases from the community setting and in children 1–10 years old.

Additionally, a logistic regression analysis was carried out in order to confirm the descriptive observations and to estimate the odds of hRSV detection according to some individual factors (such as sex and age) or healthcare setting (Figure 1). While sex was not associated to hRSV infection, significant statistical differences in risk were observed by age, with the highest probabilities of infection documented among babies under 5 years old and, particularly, in the age-group 12–23 months (OR = 4.20, *p*-value < 0.001; age-group 5–10 years as a reference).

As regards the healthcare setting, a two-fold risk of positivity to hRSV (adj-OR = 2.29, *p*-value < 0.001) was highlighted in outpatients from the community setting as compared to the cases admitted to hospitals.

Table 2 reports the distribution of relevant pre-existing diseases documented in the study population according to hRSV positivity. Nearly 6% of patients who tested positive for hRSV had at least one comorbidity, although no difference was found in infection rates between subjects with and without underlying medical conditions (data not shown).

In this regard, it is noteworthy how in the majority of cases with comorbidities hRSV-B was more represented, either in total (60.9%) or regardless the single medical condition considered, and the difference was even more pronounced for specific categories of patients, such as diabetics (72.0%), obese (77.8%), and cancer patients (78.6%).

Occurrence of complications affecting the respiratory system, including respiratory failure, ARDS, and pneumonia, was documented in about 10% of patients, considering the whole set of individuals with available data; although hRSV in these complicated patients was detected at low prevalence, ranging from 3.1% to 5.4%, at least three-quarters of them had an hRSV-B infection.

Lastly, Figure 2 depicts the seasonal distribution of hRSV-A and hRSV-B detected between 2015–2016 and 2019–2020. Given the overall prevalence of 8.1%, hRSV rates varied across seasons ranging from 5.9% (season 2017–2018) to 9.5% (season 2016–2017), while within each season it reached the peak in the winter period (cold season).

Moreover, the proportion of hRSV subgroups varied throughout the study period; hRSV-B circulated to a greater extent during all seasons except for the last season 2019–2020, when hRSV-A prevailed over hRSV-B.

## 4. Discussion

In this study we collected data of 9584 subjects of any age presenting ILI symptoms or SARI. In general, the hRSV prevalence was of 8.1%, with a positivity rate varying by winter season and ranging between 5.9% and 9.5%. This evidence from a Southern temperate area of Italy documented a lower prevalence than the one reported in the Northern part of the country [19], where an overall hRSV prevalence of 12.9% was highlighted among ILI subjects of any age over a 4-year period, partially overlapping the 5-year seasons considered in this study. Moreover, our findings confirm a higher hRSV prevalence in Italian subjects with ILI than in other European countries such as Portugal [30], where a 2.9% hRSV positivity was reported in this group of patients.

In our series, gender resulted in not being a risk factor for hRSV infection; a similar result was reported by Pellegrinelli et al. [31] in children up to 5 years of age presenting ILI symptoms, whereas Grunberg et al. [32] highlighted a significant higher prevalence of hRSV infections in females. Nevertheless, there is still ambiguity in the global epidemiological data regarding gender [33], and sex differences in respiratory viral pathogenesis have also been hypothesized, with males more susceptible to severe outcomes from respiratory viral infections at younger and older ages [34].

Within the age groups, distinct differences were noted in the overall viral positivity. The hRSV predominated in children under 5 years of age, with the highest rate of infection between 12 and 23 months. This was consistent with previously published data [19,35,36,37] corroborating the leading role of hRSV in respiratory infections among younger children. However, in our series, nearly 5% were patients ≥50 years of age, thus exhibiting the impact of this virus also in adults and the elderly [38,39], which may consequently represent important targets for vaccination [40].

Surveillance studies assessing the molecular epidemiology of hRSV epidemics have demonstrated that hRSV subgroups may co-circulate or one may predominate over the other [41,42,43,44,45]. In our setting, hRSV-B prevailed on hRSV-A in four of the five seasons considered, while in 2019–2020 the relative proportion of hRSV subgroups totally reversed. These findings were coherent with the ones reported in the northern area of Italy between the winter seasons 2014–2015 and 2017–2018 [19], which followed a 5-year period marked by the predominance of hRSV-A [20].

Of interest, the hRSV-B was more commonly detected among adult individuals ≥35 years old as well as in the hospital setting, especially among patients presenting more severe clinical manifestations. Furthermore, in support of this finding, respiratory complications mostly correlated with this subgroup, suggesting a possible role in the pathogenesis and virulence. Unfortunately, to the best of our knowledge, there are no published studies focusing on adults, which could confirm this hypothesis. Anyway, it is useful to recall that other epidemiological investigations reported conflicting results on the severity profiles due to subgroup A or B viruses [46,47,48].

Lastly, the different viral spreads among individuals with underlying medical conditions is noteworthy. In this group of subjects, as supported by our findings, a significantly higher frequency of hRSV-B cases was found in comparison to their viral counterpart. Moreover, this observation was confirmed not only for the whole group, but also for every single underlying medical condition considered.

Modelling studies have shown greater rates of hospitalization due to hRSV in adults with high-risk conditions [49,50]. In this context, Prasad et al. [38] in New Zealand and Branche et al. [39] in USA reported significantly higher hRSV hospitalization rates among adults with chronic diseases affecting the cardiovascular system, or chronic obstructive pulmonary disease, diabetes, or end-stage renal disease, as compared with adults of similar age without each corresponding condition. Nevertheless, to the best of our knowledge, there are still no data that recognized any potential difference in the prevalence of specific hRSV subgroups and the presence of underlying medical conditions.

Our study has a number of limitations that are needed to be considered when interpreting the results. Firstly, apart from the limits related to the cross-sectional design, sampling was mostly referred to the influenza surveillance season, which in the northern hemisphere conventionally spans the period from October to April of each year. However, due to the typical distribution of hRSV epidemic peak in this temperate hemisphere [9,51], the use of ILI/SARI sentinel and non-sentinel surveillance systems may represent an adequate proxy of how much this virus is involved in their occurrence. Nevertheless, it must be kept in mind that here we reported only a selected subset of ILI in Sicily that was preventively pre-sampled by collaborating sentinel physicians.

Secondly, it should be taken into account that published data on prevalence may reflect a range of factors including the age-distribution, the target population, the study setting, and as already mentioned, the climate of the geographic area where the study was carried out [51,52,53]. This variability may justify the wide heterogeneity of reported data and, therefore, a direct comparison between different studies conducted in other countries may suffer from this limitation as well [32,54,55].

In conclusion, as we have learned from influenza before, the molecular surveillance represents a key instrument to improve the knowledge on the burden of this viral pathogen in different healthcare settings and to identify high-risk groups that may benefit from vaccine-based preventive measures [17,56]. In particular, there is a growing interest on the role of hRSV among subjects of all ages and recent research has added new evidence that individuals with comorbidities may represent a target population for hRSV vaccines [38,57].

Moreover, the use of an epidemiologic integrated approach has to be highlighted whereas the widest use of non-pharmaceutical interventions (NPIs) during the current COVID-19 pandemic has changed the epidemiologic scenario of influenza and other respiratory viruses [58,59], with a decline of influenza and hRSV detection followed by a resurgence of hRSV incidence in hospitalised children after the relaxation of public health measures [60,61]. Therefore, the importance of molecular epidemiology, in support of the surveillance systems in place against respiratory pathogens both in community and hospital settings, must be emphasised also in the light of a changing and more complex global public health scenario and of the increasing need to update vaccination strategies [62,63].

While further studies on this topic would be beneficial to confirm or refute our findings, this study adds new useful data on the epidemiology of hRSV in Italy, illustrating the importance of this pathogen as a potential cause of serious respiratory illness affecting not only the paediatric population but the elderly also.

## Figures and Tables

**Figure 1 vaccines-09-01334-f001:**
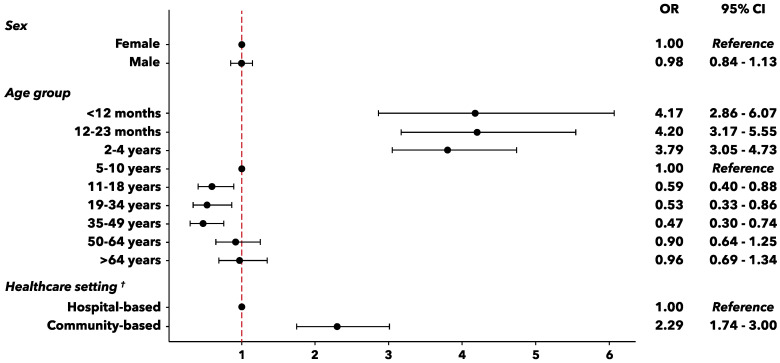
Forest plot for individual factors potentially associated to hRSV infection. OR = odds ratio, ^†^ Age-adjusted OR.

**Figure 2 vaccines-09-01334-f002:**
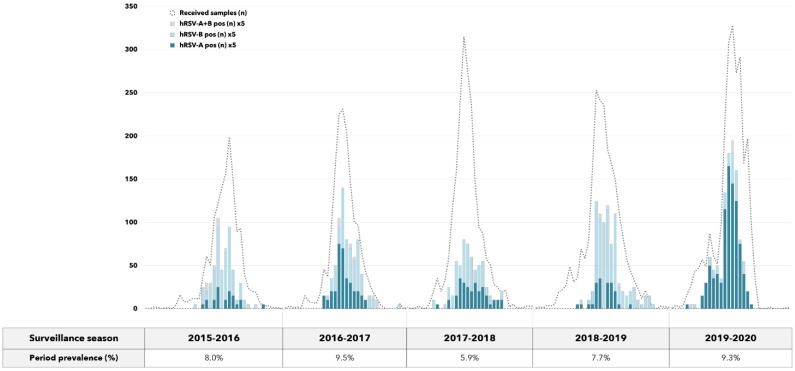
Seasonal distribution of hRSV-A and hRSV-B detected in patients with acute respiratory infections between 2015–2016 and 2019–2020. For purely graphical reasons, inter-seasonal periods were removed and the number of hRSV-positive samples multiplied by a factor of 5.

**Table 1 vaccines-09-01334-t001:** Demographic characteristics of the study population according to hRSV detection. Period: 2015–2020.

Demographic Characteristics	Total	hRSV-Positive			
		Overall	hRSV A	hRSV B	hRSV A + B
Study population [*n* (%)]	9584	772 (8.1)	359 (46.5)	395 (51.2)	18 (2.3)
Sex [*n* (%)]					
Female	4673	380 (8.1)	178 (46.8)	193 (50.8)	9 (2.4)
Male	4911	392 (8.0)	181 (46.2)	202 (51.5)	9 (2.3)
Age [years; median (IQR)]	10 (41)	3 (8)	3 (5) *	4 (11) *	3 (3)
Age-group [years; *n* (%)], *n* = 9558					
≤11 months	232	44 (19.0)	25 (56.8)	19 (43.2)	0
12–23 months	561	107 (19.1)	54 (50.5)	51 (47.6)	2 (1.9)
2–4	1816	319 (17.6)	160 (50.2)	149 (46.7)	10 (3.1)
5–10	2258	120 (5.3)	51 (42.5)	63 (52.5)	6 (5.0)
11–18	964	31 (3.2)	15 (48.4)	16 (51.6)	0
19–34	722	21 (2.9)	12 (57.1)	9 (42.9)	0
35–49	888	23 (2.6)	8 (34.8)	15 (65.2)	0
50–64	1064	51 (4.8)	18 (35.3)	33 (64.7)	0
≥65	1053	54 (5.1)	16 (29.6)	38 (70.4)	0
Healthcare setting [*n* (%)]					
Community-based	8123	713 (8.8) **	341 (47.8)	354 (49.7)	18 (2.5)
Hospital-based	1461	59 (4.0) **	18 (30.5)	41 (69.5)	0
non-ICU wards	1026	45 (4.4)	15 (33.3)	30 (66.7)	0
ICU	435	14 (3.2)	3 (21.4)	11 (78.6)	0

IQR = interquartile range, ICU = Intensive Care Unit. * *p* < 0.001, Wilcoxon rank-sum test. ** *p*-value < 0.001, Pearson’s chisquare test.

**Table 2 vaccines-09-01334-t002:** Clinical characteristics of the study population according to hRSV detection. Period: 2015–2020.

Clinical Characteristics	Total	hRSV-Positive			
		Overall	hRSV A	hRSV B	hRSV A + B
Underlying medical conditions [*n* (%)], *n* = 9509 ^†^					
At least one comorbidity	2364	138 (5.8)	51 (36.9)	84 (60.9)	3 (2.2)
Cardiovascular disease	1189	64 (5.4)	22 (34.4)	42 (65.6)	0
Lung disease	870	64 (7.4)	26 (40.6)	35 (54.7)	3 (4.7)
Diabetes	506	25 (4.9)	7 (28.0)	18 (72.0)	0
Obesity	474	18 (3.8)	4 (22.2)	14 (77.8)	0
BMI 30–40	387	17 (4.4)	4 (16.7)	13 (83.3)	0
BMI >40	87	1 (1.1)	0	1 (100.0)	0
Metabolic disorders	304	14 (4.6)	4 (28.6)	10 (71.4)	0
Cancer	219	14 (6.4)	3 (21.4)	11 (78.6)	0
Immunological disorders	171	12 (7.0)	4 (33.3)	8 (66.7)	0
Genetic disorders	64	7 (10.9)	2 (28.6)	5 (71.4)	0
Renal disease	276	5 (1.8)	2 (40.0)	3 (60.0)	0
Neurological disease	76	1 (1.3)	1 (100.0)	0	0
Liver disease	86	4 (4.6)	2 (50.0)	2 (50.0)	0
Pancreatic disease	14	1 (7.1)	1 (100.0)	0	0
Respiratory complications [*n* (%)], *n* = 9411 ^†^					
Respiratory failure	162	5 (3.1)	0	5 (100.0)	0
ARDS	242	12 (5.0)	3 (25.0)	9 (75.0)	0
Pneumonia	552	30 (5.4)	7 (23.3)	23 (76.7)	0

ARDS = Acute Respiratory Distress Syndrome. **^†^** Percentages are not mutually exclusive.

## Data Availability

Data available upon request.

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
