# Peer review of "Respiratory Syncytial Virus: New Challenges for Molecular Epidemiology Surveillance and Vaccination Strategy in Patients with ILI/SARI"

_vaccines, 2021, doi:10.3390/vaccines9111334_

Round 1
Reviewer 1 Report
The authors report on the epidemiologic trend of RSV over 5 respiratory seasons in Sicily, Italy. Age, viral subtype and setting (eg. community v. hospital) and limited clinical data (eg presence of comorbidities) are described.
Overall, the findings are well presented. There are innumerable studies on this topic, perhaps the distinctive feature here is the contemporary description of RSV activity in this region.
Introduction:
- Overall the pertinent concepts are present. It should be mentioned that RSV is the main etiology of bronchiolitis, a unique clinical entity that affects children (usually <2y) often times causing hospitalization, and that this would be reflected in the results with a majority of subjects seeking medical care in this young age group.
- This is the only section that would need moderate editing. Some sentences are too long and hard to follow, commas are missing. The structuring may be following a literal translation. It was not an easy read.
Methods and Inclusion criteria:
- The authors should describe the study population better. Who is enrolled/included? Is this a convenience sample? Also, since the study period consists of consecutive influenza (winter) respiratory seasons, the authors should define this in the manuscript (e.g. October - April) instead of referencing to #26, where it is also difficult to follow.
Results:
- Do the authors have any data regarding co-infection with other respiratory viruses (besides RSV-A + RSV-B coinfection)?
Discussion:
- Can RSV be present outside of the 'winter' seasons in this area in Italy? The authors described before influenza prevalence of ~40% during this time, RSV may be displaced to other months. Also, could that also affect serotype distribution?
- Could the authors propose an explanation for the increased representation of hRSV-B in hospitalized individuals? Did that observation hold during the 2019-2020 season when hRSV-A predominated?
References:
- Recommend to avoid unnecessary ones, for example the last four (62-65).
- After a Pubmed search, I suggest the authors include a recent similar study in their discussion of RSV epidemiology in children in Italy (Azzari C et al, Ital J Pediatr 2021;47:198).
Reviewer 2 Report
Τhis is an interesting manuscript. Epidemiological surveillance of clinically important viral infections is the cornestone for an optimal management of patients. Therefore, despite the limitations of study design, already mentioned in your article, is a topic of interest for the readers and is suitable for publication.
Reviewer 3 Report
I've read with great interest the present paper on the epidemiology of hRSV in the Italian region of Sicily, a main Mediterranean Island with nearly 4.8 million inhabitants (of them, 10% < 11 y.o. according to available demographics).
Authors from the Regional center of Palermo suggest that around 8.1% of all ILI/SARI reported to their laboratory during the time period 2015-2020 were in fact associated with hRSV infection. The risk was way greater in younger age groups (i.e. OR 4.17 and 4.20 for < 11 months and 12 - 23 month, respectively, when compared to the reference group of 5-10 years).
The study is substantially well reported, and the results sustantiated by appropriated references. However, in my opinion, some improvements are still required. More precisely:
- According to the Authors, "most of the participating subjects (84.8%...) were outpatients accessing community-based services and managed for ILI symptoms by sentinel physicians ...". Even though this sampling strategy is both quite common and useful (i.e. it increases the recruitment of individuals from the community settings), it also may bear the risk of the oversampling of individuals with certain clinical characteristics. I would suggest the authors to implement in the methods some snapshots on the working definition of ILI that was preventively shared across the sentinel physicians in order to activate molecular epidemiology surveillance, while this factor should be implemented across the potential limits of the study (i.e. you're not reporting on the whole of ILI from Sicily, but rather from a selected subsect of ILI that was preventively pre-sampled by the sentinels).
- It would be particularly interesting to report (Table 1) not only the incidence across age groups, but also by calender year, in order to appreciate whether any trend in both occurrence and potential complication did occur (this information would not duplicate that from Figure 1).
- I've a minor and substantially formal concern on Table 1; more precisely the column with the OR estimates is (at least in my opinion) somewhat inconsistent with the design of the previous sections of the very same table (in fact, percentual values are calculated on the rows, while the OR are substantially referring to the values by column). It would be possible convert the data about the OR estimates in a separated Forrest plot, in order to overcome this potential inconsistency?
